# Cardiac Magnetic Resonance and Cardiac Implantable Electronic Devices: Are They Truly Still “Enemies”?

**DOI:** 10.3390/medicina60040522

**Published:** 2024-03-22

**Authors:** Marco Fogante, Giovanni Volpato, Paolo Esposto Pirani, Fatjon Cela, Paolo Compagnucci, Yari Valeri, Adelina Selimi, Michele Alfieri, Leonardo Brugiatelli, Sara Belleggia, Francesca Coraducci, Giulio Argalia, Michela Casella, Antonio Dello Russo, Nicolò Schicchi

**Affiliations:** 1Maternal-Child, Senological, Cardiological Radiology and Outpatient Ultrasound, Department of Radiological Sciences, University Hospital of Marche, 60126 Ancona, Italy; paolo.espostopirani@ospedaliriuniti.marche.it (P.E.P.); fatjon.cela@ospedaliriuniti.marche.it (F.C.); giulio.argalia@ospedaliriuniti.marche.it (G.A.); 2Cardiology and Arrhythmology Clinic, University Hospital “Azienda Ospedaliero-Universitaria delle Marche”, 60126 Ancona, Italy; giovanni.volpato@ospedaliriuniti.marche.it (G.V.); paolocompagnucci1@gmail.com (P.C.); yarivaleri1@gmail.com (Y.V.); selimi.adelina1@gmail.com (A.S.); m.alfieri95@gmail.com (M.A.); s1102285@pm.univpm.it (L.B.); sarabelleggia94@gmail.com (S.B.); francescacorad@gmail.com (F.C.); michela.casella@ospedaliriuniti.marche.it (M.C.); antonio.dellorusso@ospedaliriuniti.marche.it (A.D.R.); 3Department of Clinical, Special and Dental Sciences, Marche Polytechnic University, 60121 Ancona, Italy; 4Department of Biomedical Sciences and Public Health, Marche Polytechnic University, 60121 Ancona, Italy; 5Cardiovascular Radiological Diagnostics, Department of Radiological Sciences, University Hospital of Marche, 60126 Ancona, Italy; nicolo.schicchi@ospedaliriuniti.marche.it

**Keywords:** cardiac magnetic resonance, cardiac implantable electronic device, pacemaker, implantable cardiac defibrillator, loop recorder, safety, artifacts

## Abstract

The application of cardiac magnetic resonance (CMR) imaging in clinical practice has grown due to technological advancements and expanded clinical indications, highlighting its superior capabilities when compared to echocardiography for the assessment of myocardial tissue. Similarly, the utilization of implantable cardiac electronic devices (CIEDs) has significantly increased in cardiac arrhythmia management, and the requirements of CMR examinations in patients with CIEDs has become more common. However, this type of exam often presents challenges due to safety concerns and image artifacts. Until a few years ago, the presence of CIED was considered an absolute contraindication to CMR. To address these challenges, various technical improvements in CIED technology, like the reduction of the ferromagnetic components, and in CMR examinations, such as the introduction of new sequences, have been developed. Moreover, a rigorous protocol involving multidisciplinary collaboration is recommended for safe CMR examinations in patients with CIEDs, emphasizing risk assessment, careful monitoring during CMR, and post-scan device evaluation. Alternative methods to CMR, such as computed tomography coronary angiography with tissue characterization techniques like dual-energy and photon-counting, offer alternative potential solutions, although their diagnostic accuracy and availability do limit their use. Despite technological advancements, close collaboration and specialized staff training remain crucial for obtaining safe diagnostic CMR images in patients with CIEDs, thus justifying the presence of specialized centers that are equipped to handle these type of exams.

## 1. Introduction

The use of cardiac magnetic resonance (CMR) in clinical practice has significantly increased in recent years due to technological advancements, increased availability, and expanded clinical indications [1,2,3]. Compared to echocardiography, CMR allows for the identification of myocardial tissue alterations, such as edema and fibrosis [4,5,6]. Additionally, it represents the gold standard method for evaluating cardiac morpho-functionality. To achieve this purpose, cine steady-state free procession (SSFP) sequences, T1-weighted and T2-weighted black-blook (T1-BB and T2-BB) sequences, T1 and T2 mapping, and late gadolinium enhancement (LGE) sequences are used. Identifying these alterations is essential for the diagnosis, prognosis, and management of major cardiac pathologies such as ischemic heart disease, non-ischemic cardiomyopathies, and myocarditis [7,8,9].

Meanwhile, in the field of cardiology, there has been a rapid increase in the use of implantable electronic devices (CIEDs) such as pacemakers (PKMs), implantable cardiac defibrillators (ICDs), and implantable loop-recorders (ILRs). These devices have significantly contributed to improvements in diagnosing and treating patients with cardiac rhythm disorders, consequently reducing morbidity and mortality [10,11]. Additionally, since one of the main causes of arrhythmia is cardiac fibrosis, it is estimated that 50–70% of patients with CIEDs require CMR during their lifetime. CMR imaging can be used in patients with CIED in various clinical scenarios, such as evaluating the causes of heart failure, studying the etiology of cardiomyopathies, and in cases of acute inflammatory conditions such as myo-pericarditis. Indeed, when compared to other imaging techniques such as computed tomography, this method provides much greater diagnostic accuracy through a high signal-to-noise ratio and contrast-to-noise ratio [12,13].

Until a few years ago, the presence of CIEDs was considered an absolute contraindication to performing CMR due to the significant safety concerns related to the interaction between the magnetic field and the electronic device. Moreover, they caused significant artifacts in the MRI image. However, in recent years, technological innovation has greatly reduced these issues. Nevertheless, there are still limitations in performing these exams, and while an increasing number of patients with CIEDs require CMR, the presence of these devices often raises concerns and skepticism among radiologists and cardiologists when performing CMR. Therefore, this topic is frequently debated within the scientific community [14,15].

This work aims to provide a brief overview of CRM and CIED, as well as to review the scientific literature, identifying issues related to the interaction of CIEDs with CMR, both in terms of safety and image degradation, providing possible solutions, proposing a protocol for the management and execution of these exams, and suggesting alternative radiological methods for cardiac tissue characterization.

## 2. Brief Overview of Cardiac Magnetic Resonance

CMR plays a leading role in the morpho-functional and tissue evaluation of the heart. Cine SSFP sequences are T1/T2-weighted sequences that use short repetition and echo times. They allow for the evaluation of wall thickness, cardiac mass, chamber volumes, and segmental and global contractile functions. In addition, they enable the anatomical and functional evaluation of heart valves, identifying pathological alterations such as insufficiency and stenosis [16,17].

Besides morpho-functional assessment, cardiac tissue characterization represents the true added value of CMR comparatively to echocardiography. Sequences used for this purpose include T1-BB, T2-BB, T1 mapping, T2 mapping, and LGE sequences. T1-BB and T2-BB sequences are inversion recovery sequences that allow for the anatomical evaluation and identification of myocardial edema, which are present in cases of myo-pericarditis. T1 and T2 mapping sequences allow for the calculation of native T1 and T2 values, identifying diffuse fibrosis, adipose infiltration, and myocardial edema occurring in numerous pathologies such as sarcoidosis, amyloidosis, cardiomyopathies, myocarditis, and Anderson–Fabry disease [18,19,20].

The LGE sequence is one of the most important sequences in CMR. It consists of inversion recovery T1-weighted gradient echo sequences, performed 8–10 min after contrast agent administration, allowing the identification of focal fibrosis, which may be a consequence of ischemic heart disease, or may be a part of the pathological process of numerous cardiomyopathies, such as hypertrophic cardiomyopathy, dilated cardiomyopathy, infiltrative cardiomyopathy, and arrhythmogenic cardiomyopathy, or may be a result of inflammatory pathologies, such as myocarditis [21,22,23,24].

Understanding the tissue alterations of cardiac pathologies and the possibility of visualization via non-invasive CMR imaging allows, on the one hand, to reduce the number of cardiac biopsies for diagnosis and, on the other hand, better patient management, paving the way for personalized treatments [5,25,26].

## 3. Brief Overview of Cardiac Implantable Electronic Devices

PKMs are CIEDs that regulate or restore cardiac activity by generating electrical impulses with durations ranging from 0.5 to 2.5 milliseconds and voltages ranging from 0.1 to 15 volts, occurring up to 300 times per minute. PKMs are equipped with pulse generators, electrocatheters, and sensing circuits. The electrodes are positioned in the cephalic vein or subclavian vein via an incision under the clavicle, and are subsequently connected to the generator implanted in a pre-pectoral pocket. 

PMKs are classified as single-chambered and dual-chambered. Single-chambered PKMs stimulate the atrium or ventricle, while dual-chambered PKMs coordinate the activity of both cardiac chambers. There are also PKMs that involve placing electrodes in the right atrium, right ventricle, and coronary sinus branches; this is to stimulate the infero-lateral wall of the left ventricle in case of cardiac resynchronization therapy (CRT). 

The two most common indications for PKM implantation are atrioventricular block (AVB) and sick sinus syndrome (SSS), which cause excessively slow heartbeats. SSS is a group of cardiac arrhythmias that includes sinus node dysfunction, sinus arrest, sinus bradycardia, and atrial tachyarrhythmias alternating with bradycardia (tachy-brady syndrome). CRT is indicated in symptomatic patients who are clinically refractory to New York Heart Association (NYHA) class III/IV heart failure therapy, with a wide QRS complex, with left bundle branch block morphology and ejection fraction falling below 35% [27,28]. 

The main complications that may occur during PKM implantation are pocket and electrocatheter infections, the displacement of the PMK, pneumothorax, hematoma formation, thromboembolism, and sepsis. 

The most significant innovations in PKMs include the introduction of X-ray identifiable generators and the use of leadless PKMs that combine a generator and electrode catheter with direct implantation into the cardiac chamber, which are both indicated in cases of problematic traditional PKM implantation, such as subclavian vein obstruction, pocket infection, and electrode catheter fracture [29,30,31]. 

Figure 1 shows a patient with a dual-chamber PMK.

An ICD is a CIED which is capable of detecting and treating irregular heart rhythms, including severe arrhythmias such as ventricular tachycardia and ventricular fibrillation. The ICD is equipped with a generator, capacitor to store and distribute shocks, microprocessor for data collection and control, and electrocatheters for sensing, stimulation, and defibrillation. 

An ICD is indicated for primary prevention in patients at risk of developing potentially fatal sustained ventricular arrhythmias, such as in cases of ischemic heart disease, hypertrophic cardiomyopathy, and long QT syndrome, or for secondary prevention in patients with previous cardiac arrest or sustained ventricular arrhythmias. Finally, ICDs may be indicated in cases of heart failure, especially with reduced ejection fraction. 

The main complications of ICDs can arise from inappropriate shocks, infections, thrombotic events, perforations, and severe bleeding. 

Subcutaneous ICDs (S-ICDs) have been developed and are frequently used in younger patients without the need of pacing, and with problematic venous access, secondary to the presence of dialysis or complex cardiac architecture, as well as in cases of previous device infection. Moreover, the generator is in the left-lateral thoracic wall, and is larger than transvenous ICDs [30,32]. Figure 2 and Figure 3 show patients with a dual-chamber ICD and an S-ICD, respectively.

The ILR is a rectangular-shaped CIED without wires, which allows for prolonged electrocardiogram (ECG) monitoring for up to three years. The LR contains a pair of sensing electrodes at each end and stores a bipolar ECG. The storage capacity is limited to 49 min, during which the LR records ECGs by deleting previous recordings, hence the name ILR. It can be activated automatically with a preset criterion, or by the patient themselves when symptoms occur. The ILR is inserted subcutaneously in the left parasternal region above the fourth intercostal space under local anesthesia. 

Compared to 24 h ECGs, the ILR has the advantage of identifying recurrent and infrequent cardiac arrhythmias. In fact, the PICTURE study reported that the ILR identified cardiac arrhythmias in 78% of patients with unexplained syncope [33]. Therefore, the main indications for ILR implantation are recurrent symptomatic arrhythmias, unexplained syncope, risk stratification following myocardial infarction, the evaluation of cryptogenic strokes, and the management of patients with atrial fibrillation [34,35]. 

There are different types of ILRs, and the smallest available system measures 44.8 mm × 7.2 mm. 

Possible complications include pain at the implant site, a local skin reaction, implant site infection, and device migration [36,37]. 

Figure 4 shows a patient with an ILR. 

## 4. Safety Considerations

It is necessary to know the safety considerations for MRIs in patients with CIEDs. “MRI Safe” items are non-conductive, non-metallic, and non-magnetic materials that pose no known hazards in MRI environments. Historically, CIEDs were considered an absolute contraindication to MRIs due to the presence of ferromagnetic materials. Currently, there are no PMKs or ICDs that have been declared “MRI safe” by the Food and Drug Administration. 

Since 2008, manufacturers have developed “MRI-conditional” CIEDs with a reduced ferromagnetic material, improved lead design, and specific programming modes to mitigate risks. MRI-conditional labeling requires the use of generators and leads from the same manufacturer. The manufacturer defines approved magnetic field strengths and SARs for scanning. The first “MRI conditional” CIED was approved by the FDA in 2011 [38,39]. 

All currently available CIEDs are “MRI conditional”, meaning that, under specific conditions, they are proven to pose no known hazards in the MRI. The conditions for most devices are as follows: scanning at 1.5 or 3.0 Tesla; SAR < 2.0 W/kg; gradient slew rate of <200 T/m/s; patient assistant (handheld activator) must not be taken into the MRI-controlled room (MRI unsafe); and the minimization of the sequence length and number [5,40]. 

CIEDs that do not meet MRI conditionality criteria are labeled as “MRI non-conditional” or “MRI unsafe”. Indeed, there are some absolute contraindications of MRI in patients with CIEDs. Additional non-conditional components include epicardial leads; abandoned leads; fractured leads; lead adapters or extenders; devices implanted in non-thoracic locations; a CIED system with leads from different manufactures, even if those leads have been approved as part of another MR-conditional system; and temporary transvenous PMKs, because these are more prone to spontaneous dislodgement and heating during MRI. Moreover, relative contraindications of MRI in patients with CIEDs include lead implanted in six weeks before the MRI, as well as whether the field of view of the exam overlaps with the region of the CIED. Understanding the potential effects of MRIs on CIEDs while adhering to safety guidelines is crucial to ensure the well-being of patients during these procedures [41].

## 5. Issue: Safety

Safety issues relating to CIEDs with CMR mainly arise from the device’s interaction with the magnetic field. Indeed, during CMR, there are three main magnetic fields: a static magnetic field (measured in Tesla), which in clinical practice can be 1.5 T or 3.0 T, and two dynamic magnetic fields, secondary to the activation of gradients and radiofrequency pulses [42]. 

The static magnetic field can cause CIED movement, caused by the attraction of the magnetic field forces to the device’s ferromagnetic components present in the battery and reed switch, a magnetically activated switch that allows the device to be placed in “magnet mode”. Moreover, device dislodgement is related to the strength of the magnetic field, its spatial gradient, the device’s mass and shape, and its magnetic susceptibility. Modern CIED components limit the use of ferromagnetic materials, which has reduced the risk of mechanical effects. In ICDs, the amount of ferromagnetic components is generally higher when compared to PMKs. To account for potential issues, CMR scans should be avoided for up to six weeks following implantation. However, the movements induced by these forces are negligible for magnetic fields up to 1.5 T [43].

The magnetic fields resulting from the activation of gradients and radiofrequency pulses are dynamic, and can induce electrical currents in the device. This can potentially cause irregularities in the ECG, with possible altered recordings, cancellations, or modifications of previous recordings, leading to incorrect diagnoses. 

To manage the electromagnetic effects, electrophysiologists manually reprogram CIEDs both before and after MRI scans. Reed switch activation can reprogram the PMK and ICD to a “magnet mode” when exposed to a strong magnetic field, such as during a CMR. This mode allows for the PMK to be set to asynchronous pacing (AOO, VOO, or DOO), where the device paces the ventricle at a continuously programmed heart rate, or in a sensing only mode (ODO), based on the patient’s characteristics. Theoretically, programming pacing to an asynchronous mode could be problematic because it might compete with the heart’s intrinsic rhythm; therefore, pacing is often set at a high heart rate [42]. Additionally, the reed switch allows for the suspension of anti-tachycardia therapies in ICDs. Indeed, ICDs can experience malfunctions if not properly programmed before an MRI exam, leading to the incorrect detection of ventricular arrhythmias, anti-tachycardia pacing, or shocks. In patients with ICDs, anti-tachycardia therapy is turned off during the MRI, being reactivated afterward. 

Continuous monitoring of the patient during the MRI is essential, with external defibrillation pads ready for use. If a severe brady/tachy arrhythmia occurs, the CMR examination must be interrupted, and the patient must be treated promptly. Moreover, CIED can undergo a power-on reset (POR) phenomenon. The POR acts as a protective mode, resetting the device’s default settings when it detects damage or issues, or when the battery falls below a critical level. Therefore, it is necessary to download the data stored in the CIED prior to the scan, reevaluate the recordings following the imaging process, and delete any altered recorded data [44,45].

Radiofrequency pulses cause energy absorption via tissues in the form of a specific absorption rate, measured in watts per kilogram. Device overheating is mainly due to the pulsed radiofrequency field, creating a local energy concentration via the “antenna effect”. Overheating can lead to tissue damage, increased pacing thresholds, and arrhythmias. The “antenna effect” is influenced by the duration, power, and spatial proximity of the pulsed radiofrequency field, and is more significant with epicardial electrocatheters. To minimize the heating risks, new lead designs have been developed to reduce the current induction. Moreover, abandoned electrocatheters generally carry a higher risk of heating, making MRI scans a contraindication in such cases [46,47].

Finally, the radiofrequency pulses can cause rapid depletion of the CIED’s battery. Premature battery depletion can be a concern, particularly for patients with low battery levels at the time of the MRI. This depletion can be due to increased impedance, threshold changes, or excessive sensing, but it may also be an apparent issue caused by POR. In most cases, battery depletion is transient, and normal values are restored within approximately three to six months following the CMR. However, potential risks should be evaluated before imaging, especially for pacemaker-dependent patients or those with low battery voltages [48]. Table 1 summarizes safety issues between CMR and CIED interactions.

## 6. Issue: Artifact

Artifacts on the CMR image are another issue relating to the presence of CIEDs. The extent of the artifacts cannot be predicted in advance. Several factors influence artifact formation, including device characteristics (composition, size, shape, and orientation), position, the magnetic susceptibility of the CIED, the distance from the region of interest, MRI sequences used, and magnetic field strength [49].

The larger the size of the CIED, the more pronounced the artifacts in the image. Regarding the position, CMR is particularly susceptible to CIED artifacts because it is positioned near the heart, and the closer the generator is to the apex of the left ventricle, the greater the artifacts. Additionally, the stronger the magnetic field, the greater the artifacts [50,51].

The MRI sequences most vulnerable to artifacts are cine SSFP and LGE sequences; this is because they require a homogeneous magnetic field to adequately balance gradients. Magnetic susceptibility is due to the ferromagnetic components of the CIED, which cause static magnetic field inhomogeneity, resulting in signal loss and hyperintensity artifacts [15]. According to the Heart Rhythm Society (HRS) expert consensus statement published in 2017, when using fast gradient echo (FGE) sequences for cine imaging as well as wideband sequences for LGE imaging, image quality may be improved where artifacts are present [48,52]. 

Signal loss artifacts create dark bands within 5–12 cm around the device, mainly affecting the anterior cardiac wall. To reduce this artifact, sequences such as fast gradient echo (FGE) with shorter echo and repetition times are used, which are more stable to magnetic field inhomogeneity, and can be used as alternatives to cine SSFP sequences. Other methods include using a frequency-scout method before cine imaging and acquisition planes perpendicular to the CIED [53].

Hyperintensity artifacts cause increased signal intensity in the anterior wall of the left ventricle in LGE sequences. This artifact occurs because the inversion pulse used in LGE sequences has a spectral bandwidth of approximately 1 kHz, but in the presence of a CIED, regions of the heart as far as 5–10 cm may undergo a frequency shift beyond this bandwidth (2–4 kHz), resulting in the incomplete nulling of signals and persistent hyperintensity. To address this, wider bandwidth LGE sequences (up to 3.8 kHz) have been developed to ensure the proper inversion of the myocardium and to eliminate hyperintensity artifacts [39,54,55,56]. The disadvantage of wideband LGE sequences is the increase in SAR, which is considered acceptable up to 2 W/kg. Studies have shown the effectiveness of wideband inversion recovery LGE techniques in reducing device-related hyperintense artifacts, thus resulting in improved image quality [38,57]. It is important to note that, while wideband techniques reduce some artifacts, they may not eliminate geometric distortions or signal voids caused by off-resonance. Localized shimming can help address these issues. Figure 5 and Figure 6 show examples of the CMR’s artifacts. 

Another possible solution is to use smaller CIEDs. For example, smaller LR models, such as type LINQ, produce fewer artifacts (Figure 7). 

Additionally, certain techniques can help reduce artifacts; these include scanning during inspiration to increase the distance between the heart and the cardiac device, correlating different sequences, elevating the patient’s arm on the side of the CIED generator during the MRI scan to increase the distance between the heart and the device, and right-sided CIED implantation. Furthermore, the presence of an S-ICD with a large generator placed at the lateral chest wall causes more artifacts on the cardiac imaging when compared to the dual-chamber ICD [58]. Figure 8 and Figure 9 show examples of patients with a dual-chamber ICD and an S-ICD. 

Table 2 summarizes the issues related to CMR’s artifacts with CIEDs.

## 7. Clinical Protocol for Patients with CIEDs Undergoing CMR

To perform an MRI examination in a patient with a CIED “MRI-conditional”, it is recommended to follow a rigorous protocol involving the interaction of different multidisciplinary physicians. Indeed, under the guidelines of the Heart Rhythm Society, the performance of MRIs in patients with CIEDs is a Class I (strong) recommendation, only with a standardized institutional workflow [59].

The protocol should include at least three steps to conduct a safe examination in patients with CIEDs [60,61,62].

The first step is carried out prior to the examination. The first element to evaluate is the risk/benefit ratio of the examination via an interaction between the radiologist and the colleague requesting the examination. It is always advisable to consider the possibility of an alternative method, equally capable of performing the diagnostic. Once the clinical necessity of the examination is established, a thorough analysis of the device characteristics (name, serial number, model, implantation date) is necessary. It may be useful to know the type of device, which can be achieved by performing a chest X-ray to identify specific radiopaque markers from the manufacturer. The specifications of the device must be sent to the medical physicist expert at the site where the examination will be conducted; this is to assess whether the device’s characteristics and the CIED’s conditions are compatible with the MRI used. Additionally, it is recommended to wait at least six weeks after device implantation to ensure adequate healing of the wound and to minimize the risk of movement. Before performing the MRI, the device parameters, battery status, and all stored data must be checked, and CIEDs must be switched in terms MRI modality. Furthermore, obtaining informed consent from the patient regarding the risks associated with the type of examination they are undergoing is essential. It is imperative to inform the patient that if they experience any sensations of movement from the CIED, discomfort, or heat during the examination, they should promptly alert the medical staff to immediately stop the examination [39,63].

The second step is monitoring during the CMR. It is necessary to carefully monitor the patient’s vital signs, have all resuscitation and advanced life support equipment available, and ensure that the staff are prepared and ready if necessary. Visual and verbal contact with the patient throughout the examination is necessary. The duration of the CMR should be minimized, aiming not to exceed 30 min, and it should be promptly stopped if the patient experiences dangerous symptoms or if there are alterations in the patient’s vital parameters [64]. All possible techniques to reduce the number and extent of MRI image artifacts should be used [65].

The third step occurs following the scan. Once the patient has completed the examination and left the MRI room, it is important to interrogate the CIED in order to identify any artifact-related arrhythmias that may have occurred during the MRI examination, deleting them if found. It is important to perform a second evaluation of the device to compare the parameters with the pre-MRI values, scheduling a check-up 3 and 6 months from the examination. Moreover, it should be noted that the HRS statement suggests an earlier follow-up (before 3 months) when there is a more than a 1 V capture-threshold increase, a more than 50% decrease in P-wave or R-wave amplitude, or changes in pacing or shock impedance [48]. By integrating these points into the protocol, it could be possible to further improve the safety and effectiveness of MRI examinations in patients with “MRI conditional” implantable devices. Moreover, this protocol could also be applied in other cases where there may be safety concerns for patients with CIEDs, such as those undergoing radiotherapy (RT) for malignancies. In these cases, it is advisable to perform a complete evaluation of the CIED before the RT, minimize any direct exposure of the CIED to radiation, monitor the patient during RT, check the CIED at the conclusion of treatment, and schedule follow-up appointments 1 and 6 months after the treatment [31,60,61,62]. Figure 10 summarizes the proposed protocol for MRI examinations in patients with CIED.

## 8. Alternative Imaging Modalities to CMR

Computed tomography coronary angiography (CCTA) can be considered as an alternative method to CMR for the morpho-functional evaluation and tissue characterization of the myocardium in patients with CIEDs [65]. Using a retrospective protocol, CCTA can assess the wall thickness, dimensions of the cardiac chambers, volume, and ventricular contractile function. This allows for the anatomical evaluation of the heart valves, in addition to its well-known coronary assessment capability [66]. In recent years, technological advances have introduced CT techniques into clinical practice that enable tissue characterization, including dual-energy CT (DECT) and photon-counting (PC) techniques [67]. 

DECT involves images acquisition at two different kilovoltages, allowing for the spectral separation of imaging and material identification. Specifically, it can assess cardiac perfusion using iodine maps, myocardial extracellular volume, and cardiac fibrosis via the assessment of late iodine enhancement, identifying the accumulation of iodine-based contrast medium in delayed scans, similar to what occurs in CMR [68,69,70].

The PC technique relies on the ability of CT detectors to discriminate the energy of individual photons and potentially distinguish multiple materials within the same voxel. For this reason, it can evaluate myocardial iodine perfusion maps and myocardial fibrosis due to the late accumulation of the iodine contrast medium [67,71]. The limitations of the use of these techniques are mainly due to their limited availability and the lack of adequate literature supporting their diagnostic accuracy, especially for the PC technique, although more studies are emerging in support of this method. Additionally, CT involves the use of ionizing radiation, although significant technological advances have greatly reduced radiation doses [72,73,74]. Finally, it must be considered that the presence of CIEDs may induce image artifacts, such as photon starvation/beam hardening, which can reduce the late iodine enhancement detection [65].

## 9. Conclusions

In conclusion, technological advances in CMR and CIEDs have, under certain conditions and with appropriate protocols, significantly improved patient safety, as well as allowed for the obtaining of diagnostic CMR images. Although technological development will enable a greater use of alternative methods for MRI, the increase in the number of patients with CIEDs will lead to an increase in the number of CMR examinations in the presence of implantable devices. Close collaboration among various specialized figures, and the adequate preparation of both medical and technical staff in managing the examination and any issues that arise are necessary when performing these examinations. In addition, it may be useful to limit this type of examination to certain centers where dedicated and adequately trained personnel are available. These strategies collectively aim to enhance the safety of MRI procedures for patients with CIEDs, while providing medical professionals with the appropriate tools and guidelines to minimize potential risks.

## Figures and Tables

**Figure 1 medicina-60-00522-f001:**
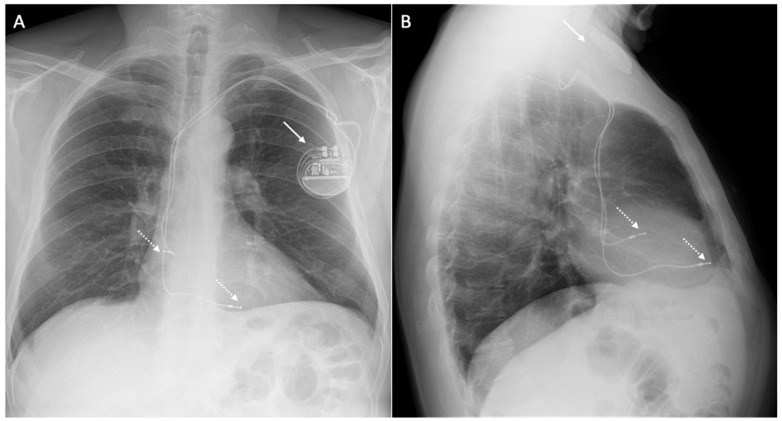
X-ray of a 68-year-old patient with a dual-chamber pacemaker. The figure shows the chest X-ray of a patient with a dual-chamber pacemaker in posteroanterior (**A**) and lateral (**B**) views, with the generator located in the left pre-pectoral pocket (white arrow) and electrocatheters located in the right atrial appendage and right ventricle apex (dotted arrow).

**Figure 2 medicina-60-00522-f002:**
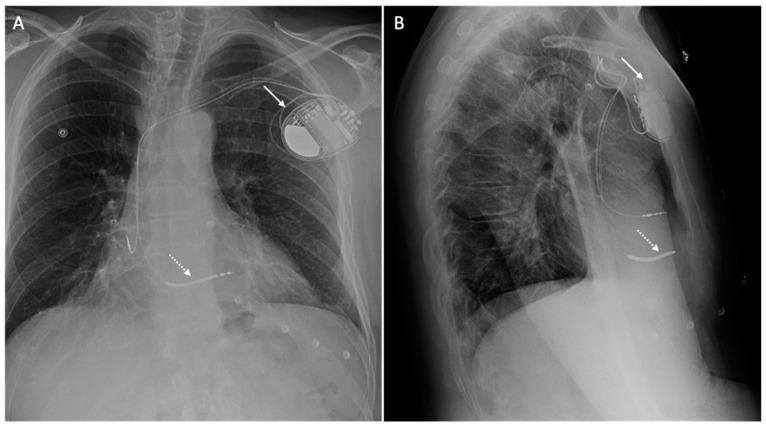
An X-ray of a 73-year-old patient with a dual-chamber implantable cardiac defibrillator. The figure shows the chest X-ray of a patient with a dual-chamber implantable cardiac defibrillator in posteroanterior (**A**) and lateral (**B**) views, with the generator located in the left pre-pectoral pocket (white arrow) and the defibrillator electrocatheter located in the right ventricle (dotted arrow).

**Figure 3 medicina-60-00522-f003:**
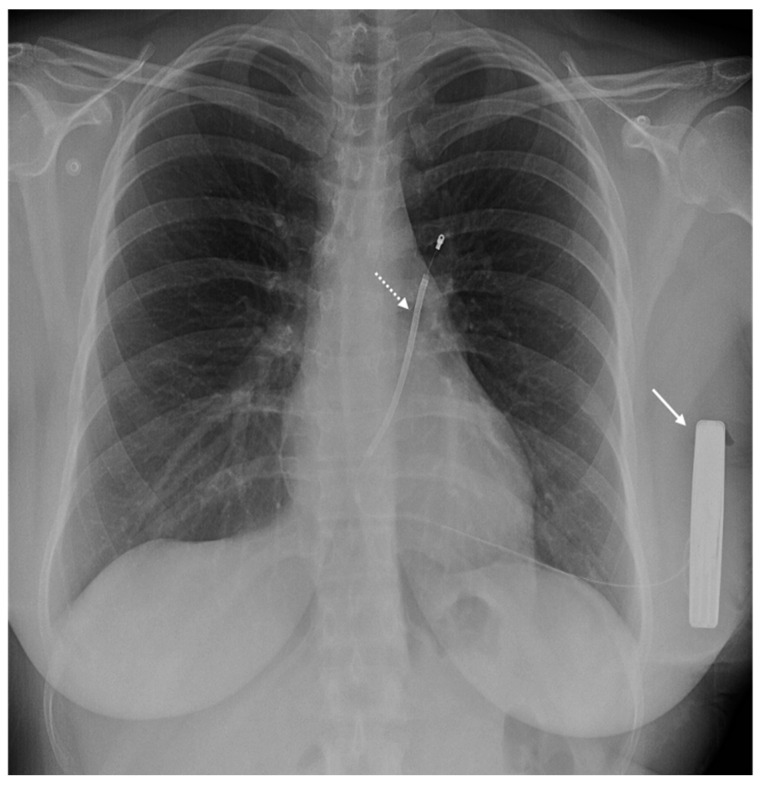
An X-ray of a 65-year-old patient with a subcutaneous implantable cardiac defibrillator. The figure shows the chest X-ray of a patient with a subcutaneous implantable cardiac defibrillator in a posteroanterior view, with the large generator located in the left lateral wall (white arrow) and the subcutaneous defibrillator electrocatheter located in the left parasternal position (dotted arrow).

**Figure 4 medicina-60-00522-f004:**
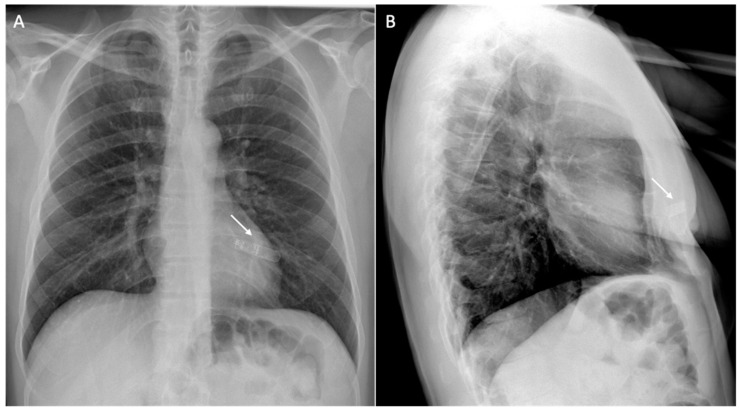
An X-ray of a 60-year-old patient with a loop recorder. Figure 4 shows the chest X-ray of a patient in posteroanterior (**A**) and lateral (**B**) views, with a subcutaneous loop recorder located in the anterior chest wall (white arrow).

**Figure 5 medicina-60-00522-f005:**
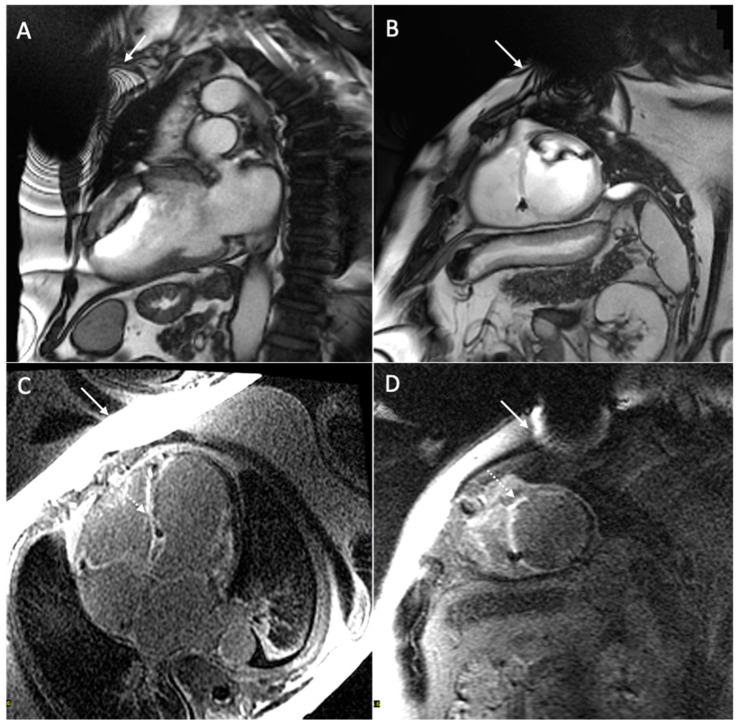
Cardiac magnetic resonance of a 68-year-old patient with a dual-chamber pacemaker. The figure shows signal loss artifacts characterized by dark bands around the device in cine steady-state free processing sequences in both long-axis (white arrow, (**A**)) and short-axis (white arrow, (**B**)) views, and hyperintensity artifacts in late gadolinium enhancement sequences in four-chamber (white arrow, (**C**)) and short-axis (white arrow, (**D**)) views. The presence of hyperintensity artifacts may make it difficult to distinguish the diffuse transmural septal and right ventricular late gadolinium enhancement (dotted arrow, (**C**,**D**)).

**Figure 6 medicina-60-00522-f006:**
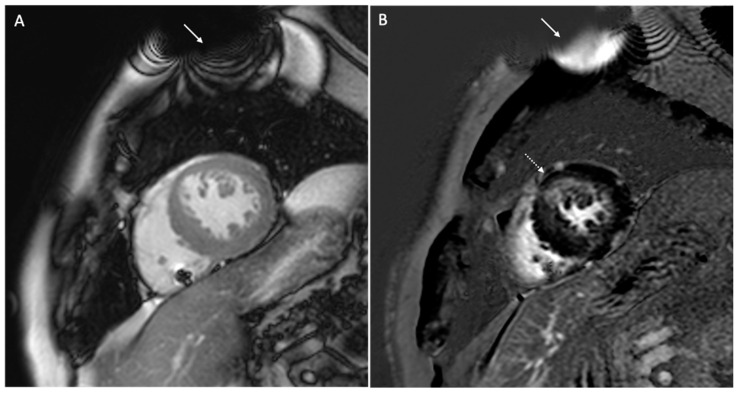
Cardiac magnetic resonance of a 67-year-old patient with a dual-chamber pacemaker. The figure shows signal loss artifacts characterized by dark bands around the device in cine steady-state free processing sequences (white arrow, (**A**)) and hyperintensity artifacts in late gadolinium enhancement sequences (white arrow, (**B**)), both in a short-axis view. Both artifacts do not involve the myocardium because the generator is smaller and located in a lateral subpectoral pocket, allowing for the detection of subendocardial circumferential late gadolinium enhancement (dotted arrow).

**Figure 7 medicina-60-00522-f007:**
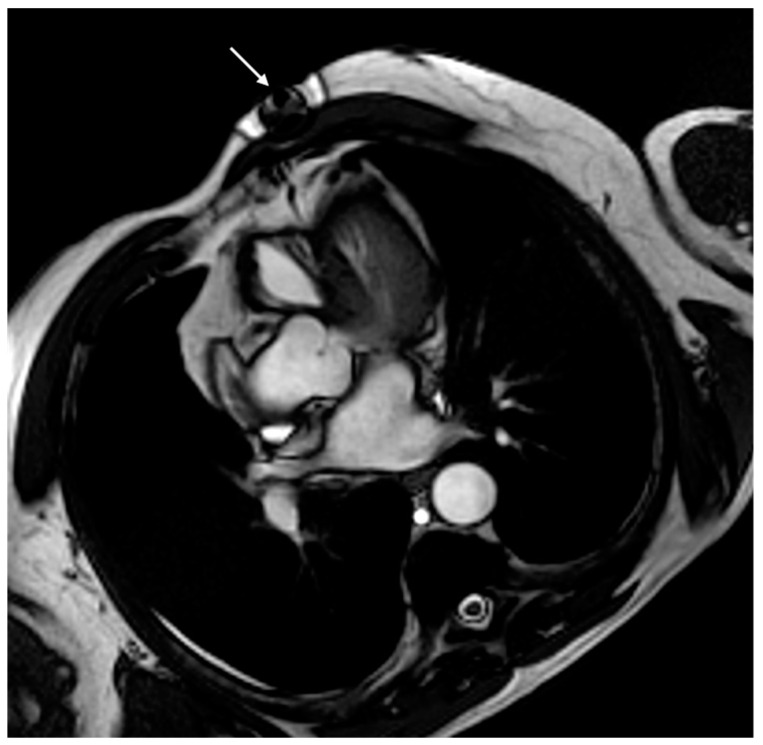
CMR of a 65-year-old patient with a loop-recorder. The figure shows signal loss artifacts (white arrow) in cine SSFP sequences, not involving the anterior cardiac wall due to the small size of the loop-recorder.

**Figure 8 medicina-60-00522-f008:**
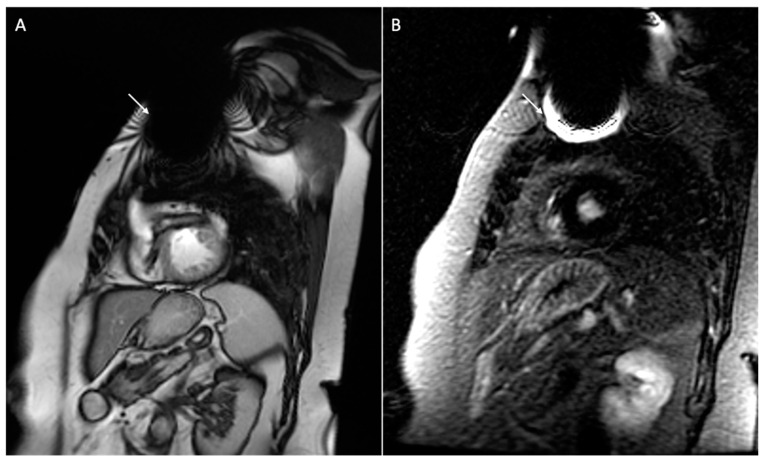
Cardiac magnetic resonance of a 71-year-old patient with a dual-chamber implantable cardiac device. The figure shows signal loss artifacts characterized by dark bands around the device in cine steady-state free processing sequences involving the anterior cardiac wall in a short-axis view (white arrow, (**A**)), and hyperintensity artifacts in late gadolinium enhancement sequence not involving the myocardium in a short-axis view (white arrow, (**B**)).

**Figure 9 medicina-60-00522-f009:**
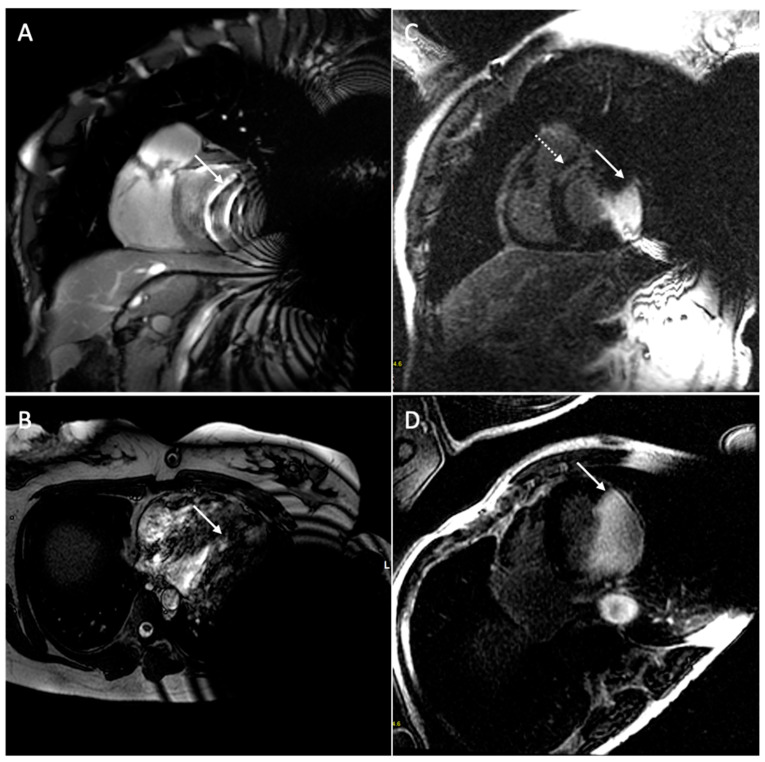
Cardiac magnetic resonance of a 75-year-old patient with a subcutaneous implantable cardiac device. Figure 9 shows signal loss artifacts characterized by dark bands around the device in cine steady-state free processing sequences in short-axis (white arrow, (**A**)) and four-chamber (white arrow, (**B**)) views, as well as hyperintensity artifacts in late gadolinium enhancement sequences in short-axis (white arrow, (**C**)) and four-chamber (white arrow, (**D**)) views. The presence of hyperintensity artifacts represents a possible differential diagnosis with intramyocardial late gadolinium enhancement at the antero-basal septum (dotted arrow, (**C**)).

**Figure 10 medicina-60-00522-f010:**
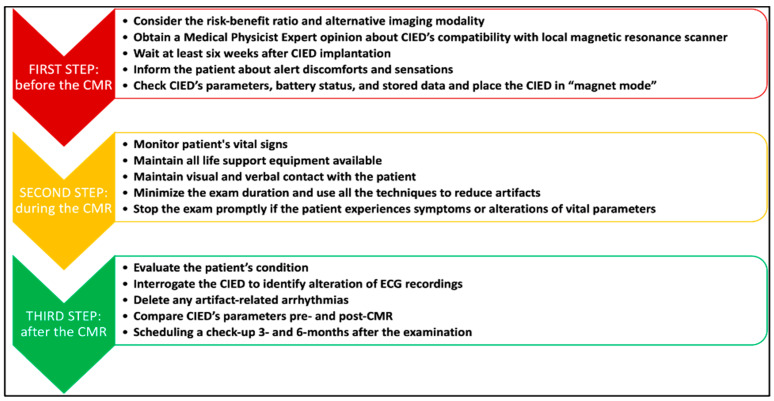
Proposed protocol for CMR exam in patient with CIED.

**Table 1 medicina-60-00522-t001:** Safety issues between cardiac magnetic resonance and cardiac implantable electronic device interactions.

Safety Issue	Cause	Risk Factors	Dangers	Minimize the Risk
Device dislodgement	MF’s attraction forces to CIED’s ferromagnetic components	MF’s strength; CIED mass, shape, and magnetic susceptibility	Cutaneous and subcutaneous lesions	Reduce CIED’s ferromagnetic components Avoided CMR up to six weeks following CIED implantation Prefer 1.5 T
ECG recordings’ modifications and deletions	Electrical currents induced by gradients and radiofrequency pulses	Duration, power, and spatial proximity of gradient field and radiofrequency field	Incorrect diagnoses, inappropriate shocks	Reed switch activation Cease ICD anti-tachycardia therapy Download the data stored before the scan, reevaluate, and delete altered recordings after the MRI
Device overheating	Local energy concentration created by the radiofrequency	Duration, power, and spatial proximity of the radiofrequency pulses Epicardial or abandoned electrocatheter	Myocardial tissue damage Increased pacing thresholds Arrhythmias	New lead designs Avoid MRI with epicardial or with abandoned lead
Premature battery depletion	Radiofrequency pulses causes increased impedance, threshold changes, and excessive sensing	Duration, power, and spatial proximity of the radiofrequency pulses Low battery before the MRI	Battery depletion	Battery depletion is usually transient POR phenomenon

Abbreviations—ECG: electrocardiogram; MF: magnetic field; CIED: cardiac implantable electronic device; MRI: magnetic resonance imaging; ICD: implantable cardiac device; POR: power-on reset.

**Table 2 medicina-60-00522-t002:** Issues related to CMR’s artifacts with CIEDs.

Types of Artifacts	Factors that Influence Artifacts	High Artifact	Low Artifact
Signal loss artifact Hyperintensity artifact	CIED’s dimension	Large device	Small device
	CIED’s position	Left-sided implantation	Right-sided implantation
	Magnetic susceptibility	High ferromagnetic component	Low ferromagnetic components
		High static MF	Low static MF
	Distance from the region of interest	Proximity to the heart	Elevate the patient’s arm
	MRI sequences used	Cine SSFP	SGE sequences
		LGE sequence with a bandwidth of about 1 kHz	LGE sequence with a wide bandwidth

Abbreviations—CIED: cardiac implantable electronic device; MF: magnetic field; SSFP: steady-state free precession; LGE: late gadolinium enhancement; SGE: spoiled gradient echo.

## Data Availability

Data are contained within the article.

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
