# Peer review of "Cardiac Magnetic Resonance and Cardiac Implantable Electronic Devices: Are They Truly Still “Enemies”?"

_medicina, 2024, doi:10.3390/medicina60040522_

Round 1

Reviewer 1 Report

Comments and Suggestions for Authors

Dear  Marco Fogante et al., 

Cardiac MRI is a valuable diagnostic tool for detecting and monitoring cardiac diseases in patients with implantable cardiac electronic devices (CIEDs). CIEDs were once considered a contraindication to CMR due to concerns of arrhythmias, lead parameter changes, and device programming issues. However, CMR can be safely performed on patients with CIEDs (conditional or not) at 1.5 Tesla if no fractured, abandoned, or epicardial leads are present. 

You have already provided a comprehensive review of CMR, including information on CIEDs, safety concerns, and artifacts that may occur during a CMR study.

My suggestions...

In section 4, regarding non-conditional MRI CIEDs, it should be noted that these devices also include CIED systems with leads from different manufacturers, even if those leads have been approved as part of another MR-conditional system.

Moreover, in section 5, regarding safety issues, it should be noted that CMR scans can be safe for most MR non-conditional CIED systems at 1.5 Tesla. However, potential risks should be discussed before imaging, especially for pacemaker-dependent patients or those with low battery voltages (2017 HRS expert consensus statement on magnetic resonance imaging and radiation exposure in patients with cardiovascular implantable electronic devices).

Additionally, it might be added in section 6, that according to the HRS expert consensus statement published in 2017, using FGE (fast gradient echo) sequences for cine imaging as well as wideband sequences for late gadolinium enhancement imaging, image quality may be improved when there are artifacts. FGE sequences improved image quality in the majority of left and right ventricle images in patients with Evera ICDs (reference: Schwitter J et al. Circ Cardiovasc Imaging 2016).

Regarding the clinical protocol (section 7), the HRS statement suggests earlier follow-up (before 3-6 months) when there is a more than 1V capture-threshold increase, a more than 50% decrease in P-wave or R-wave amplitude as well as pacing or shock impedance change.

Check the following...

1)      I believe that all chest X rays are posteroanterior views rather than anteroposterior  (gastric air bubble present)

2)       Figure 7: white arrow has been forgotten 

Author Response

Dear Reviewer, thank you very much for your time and comments . They have allowed us to improve our article.

In section 4, regarding non-conditional MRI CIEDs, it should be noted that these devices also include CIED systems with leads from different manufacturers, even if those leads have been approved as part of another MR-conditional system.

I have added it.

Moreover, in section 5, regarding safety issues, it should be noted that CMR scans can be safe for most MR non-conditional CIED systems at 1.5 Tesla. However, potential risks should be discussed before imaging, especially for pacemaker-dependent patients or those with low battery voltages (2017 HRS expert consensus statement on magnetic resonance imaging and radiation exposure in patients with cardiovascular implantable electronic devices).

I have added it.

Additionally, it might be added in section 6, that according to the HRS expert consensus statement published in 2017, using FGE (fast gradient echo) sequences for cine imaging as well as wideband sequences for late gadolinium enhancement imaging, image quality may be improved when there are artifacts. FGE sequences improved image quality in the majority of left and right ventricle images in patients with Evera ICDs (reference: Schwitter J et al. Circ Cardiovasc Imaging 2016).

I have added it.

Regarding the clinical protocol (section 7), the HRS statement suggests earlier follow-up (before 3-6 months) when there is a more than 1V capture-threshold increase, a more than 50% decrease in P-wave or R-wave amplitude as well as pacing or shock impedance change.

I have added it.

Check the following...

1)      I believe that all chest X rays are posteroanterior views rather than anteroposterior  (gastric air bubble present)

I have corrected it.

2)       Figure 7: white arrow has been forgotten

I have corrected it.

Reviewer 2 Report

Comments and Suggestions for Authors

I would like to congratulate authors for this article. But I have few suggestions .

1) It is better to include some common clinical scenarios where MRI is done for patients with CIED.  Advantages of CMR versus other imaging modalities can be discussed.

2) Absolute and relative contraindications of MRI in CIED patients should be clearly mentioned.

3) Safety of radiation therapy for various malignancies in CIED patients could have been described.

I think these will make your article complete.

Author Response

Dear Reviewer, thank you very much for your time and comments . They have allowed us to improve our article.

1)It is better to include some common clinical scenarios where MRI is done for patients with CIED.  Advantages of CMR versus other imaging modalities can be discussed.

I have added it.

2) Absolute and relative contraindications of MRI in CIED patients should be clearly mentioned.

I have mentioned it.

3) Safety of radiation therapy for various malignancies in CIED patients could have been described.

I have described it.